# Management of common minor ailments in Qatar: Community pharmacists' self-perceived competency and its predictors

**Ahmed Mohamed Makhlouf[1], Mohamed Izham Mohamed Ibrahim[1], Ahmed Awaisu[1], Saseendran Kattezhathu Vyas[2], Kazeem Babatunde Yusuff[1]***

**1** Department of Clinical Pharmacy and Practice, College of Pharmacy, QU Health, Qatar University, Doha, Qatar, **2** Wellcare Group, Doha, Qatar

* kyusuff@qu.edu.qa

**Data Availability Statement:** All relevant data are within the paper.

**Funding:** The study was funded by the Qatar National Research Foundation's Undergraduate

## Abstract

Studies focused on comprehensive assessment of self-perceived competency of community pharmacists to manage minor ailments are scanty despite that self-perceived competency is a valid determinant of task performance. The objectives of the study were to assess community pharmacists' self-perceived competency to manage fourteen common minor ailments in Qatar, and identify its significant predictors. A cross-sectional assessment of 307 community pharmacists was conducted with a pre-tested 20-item questionnaire. Self-perceived competency was assessed with nine elements on a scale of 1–10 (Maximum obtainable score: Each minor ailment = 90; each element = 140). Mann-Whitney U and bivariate logistic regression were used for data analyses. The response rate was 91.9% (282/307). The majority of the respondents were males (68.1%; 192/282), within the age range of 31–40 years (55.3%; 156/282). The minor ailments with the highest median competency score were constipation (76), and cold/catarrh (75) while travel sickness (69), and ringworm (69) had the lowest. The two condition-specific competency elements with the highest median score were recommendation of over-the-counter (OTC) medicines (115), and provision of instructions to guide its use (115). Ability to differentiate minor ailments from other medical conditions had the lowest median competency score (109). The significant predictors self-perceived competency were female gender (OR = 2.39, 95%CI: 1.34–4.25, p = 0.003), and working for chain pharmacies (OR = 2.54, 95%CI: 1.30–4.96, p = 0.006). Overall, Community pharmacists' self-perceived competency was adequate for majority of the common minor ailments, and it was highest for constipation and cold/catarrh, and specifically for the recommendation of OTC medicines and provision of instructions to guide its use. However, diagnostic ability to differentiate minor ailments from other medical conditions with similar features had the lowest median competency score. Female gender and working in chain pharmacies were the significant predictors of self-perceived competency to manage minor ailments.

Research Experience Program [UREP24- 147- 3-043]. The funder provided support in the form of honorarium for authors MII, AMM, SKV, KBY, but did not have any additional role in the study design, data collection and analysis, decision to publish, or preparation of the manuscript. The specific roles of these authors are articulated in the 'author contributions' section.

**Competing interests:** One of the authors (SKV) declared a commercial afflation to the Wellcare Group where he is currently receiving financial support in the form of monthly salaries. This does not alter our adherence to PLOS ONE policies on sharing data and materials.

## Introduction

Minor ailments are uncomplicated medical conditions that are commonly encountered in clinical practice, and are often managed with self-medication or self-care practices with or without the guidance of community pharmacists [1–3]. Despite the perceived uncomplicated nature of minor ailments, they have become a major source of clinical and financial burden especially in primary, tertiary and emergency care settings [4–7]. For example, minor ailments account for about 13% of 340 million visits to primary care physicians in the United Kingdom (UK) [7]. In addition, the estimated cost of about 136 million pounds per annum was attributed to about 8% of Emergency Department visits due to minor ailments [4–6]. However, the use of an integrated collaborative delivery model that cede the task of managing minor ailments to other primary healthcare professionals such as community pharmacists have been shown to reduce the burden associated with minor ailments [8–10]. For instance, the reported benefits include more efficient use of healthcare resources as physicians are able to focus on more serious medical conditions, reduction in patient load and waiting time, and increased patient satisfaction [4, 8, 11]. The use of these collaborative initiatives have been reported in the United Kingdom (UK), Canada, New Zealand and Australia [7, 10, 12–14]. However, an extensive literature search showed that such models of practice are rare in developing countries including Qatar.

In the State of Qatar, there has been a heavy capital investment in the public health system over the past decade with specific focus on enhancing the quality of healthcare delivery. This was encapsulated in the Qatar National Vision (QNV) 2030 that clearly defined the strategic goals in key priority areas [15]. A major pillar of the QNV 2030 is the Qatar National Health Strategy (QNHS) 2018–2022, which specified an integrated approach to the provision of a functional patient-centered primary care that is closer to home. The QNHS 2018–2022 identified community pharmacists as one of the key healthcare professionals whose active participation is crucial to expanding access to functional primary care services including the effective management of minor ailments [16].

Literature search revealed inconsistencies in information gathering and counseling practices of community pharmacists during the management of minor ailments in developing countries [17–19]. However, studies focused on comprehensive assessment of the self-perceived competency of community pharmacists to manage minor ailments are scanty. This is an essential first step, because the ability to assume a responsibility is substantially dependent on self-perceived competence to execute the tasks associated with that responsibility successfully [20, 21]. This assertion is consistent with Albert Bandura's social cognitive theory that describes the relationship between the successful completion of an assigned task and perceived self-efficacy [22]. Self-perceived competence is a major component of self-efficacy that has been identified as a social cognition construct that essentially encapsulate the ability to perform or execute a task.

Indeed, empirical evidence has shown that self-perceived competency is a valid and reliable determinant of task performance; and this is because the higher the self-perceived competency, the more confident an individual feels about executing an assigned task targeted to obtaining specific outcomes, and vice versa [23–25]. Therefore, a baseline assessment of community pharmacists' self-perceived competency to manage commonly encountered minor ailments in Qatar is crucial to determining their readiness to key into the integrated primary healthcare model of the QNHS. In addition, this may also provide new perspectives that add to global knowledge in the research area. The objectives of the current study were to conduct a baseline assessment of the self-perceived competency of community pharmacists to manage selected

minor ailments commonly encountered in Qatar, and identify the predictors of their competency level.

## Methods

### Study design

A cross-sectional assessment of community pharmacists' self-perceived competency to manage selected common minor ailments was conducted between 01 September and 30 December 2019 in the State of Qatar. This is one of the smallest countries in the Gulf of the Middle East and consists of eight municipalities with a population of 2.7 million [26].

The methodological approaches used for the study consisted of two phases. The first phase was a cross-sectional survey focused on the selection of the most commonly encountered minor ailments in Qatar, and this was used for the community pharmacists' self-competency assessment in the second phase.

### Target population and sampling

The first phase of the study involved a purposive sample of 10 physicians that was drawn at two selected healthcare facilities including the Adult Emergency Section and the Family Practice Unit of the Mobile Health Service, both of which are affiliated to Hamad Medical Corporation in Qatar. The rationale for the selection was because such facilities have been reported in published literature to devote a significant proportion of clinical resources to the management of minor ailments [4, 6].

A list of 58 minor ailments was developed after a thorough review of published literature about minor ailments [17–19, 27–31]. The 10 physicians were asked to assign a rank ranging from 1 to 10 to the minor ailments that were considered most commonly encountered in Qatar from the list of 58 minor ailments presented to them. In addition, all the 14 minor ailments ranked 9 or 10 by physicians were presented to a purposive sample of 10 community pharmacists to obtain their assessment of their suitability for inclusion in the second phase of the study, and there was no disagreement with the physicians' ranking.

The second phase of the study was a cross-sectional assessment of the self-perceived competency of a purposive sample of community pharmacists to manage the selected minor ailments in Qatar. The list of all licensed community pharmacists in Qatar was obtained from the Ministry of Public Health, and this constituted the sampling frame. Other inclusion criteria were being in practice in chain or independent pharmacies for at least one year and ability to speak and write in English language. The required number of study participants was calculated *a priori* with Raosoft® online sample size calculator and the factors used included the total number of licensed community pharmacists in Qatar (1016), alpha level (5%), confidence level (95%) and estimated response distribution of 50%. The calculated sample size was 279, but 10% was added to account for possible non-response or withdrawal, and this resulted in a final sample of 307 community pharmacists.

### Questionnaire development and structure

The community pharmacists self-perceived competency assessment was done with a 20-item questionnaire that was developed after literature review [19, 27–31]. The questionnaire was divided into two sections, including: A (community pharmacists' demographic and workload characteristics) and B (self-perceived competency to manage selected minor ailments commonly encountered in Qatar). The data collected in section A included gender, age group, nationality, community pharmacy type, duration of practice experience, highest pharmacy

degree, average consultation time for minor ailments, average number of customers per daily shift and number of customers with minor ailments per daily shift. The data gathered in section B about community pharmacists' self-perceived competency to manage selected common minor ailments in Qatar included two categories: condition-specific (9 items) and non-condition specific (3 items). The condition-specific items include description/definition, etiology, symptoms, determination of when a referral is needed, pharmacological/non-pharmacological recommendations and instruction for use, recognition of consideration for special populations and differentiation of minor ailments from other similar conditions. The non-condition-specific items were follow-up, documentation and use of information resources during management of minor ailments.

The content validity of the questionnaire was assessed by a team of three faculty members of the Qatar University's College of Pharmacy expertise and experience in the research area and a pharmacy manager in one of the community pharmacy chains in Qatar. The feedback received from the team about the relevance, validity and comprehensiveness of the items in the draft questionnaire was used to determine the 20 items included in the final questionnaire. In addition, the questionnaire was pre-tested with a convenient sample of eight community pharmacists for clarity and completeness, and the pretest data were not included in the study results. The internal consistency of the 12 items included in the final questionnaire for the assessment of self-perceived competency determined with Cronbach alpha coefficient was 0.91.

The participants ranked their responses to the 12 competency elements (condition- and non-condition specific) in Section B on a semantic differential scale of 1 to 10 (1: low competence, 10: full competence). The maximum obtainable scores for the nine condition-specific competency items for each minor ailment was 90 and 1260 for the 14 minor ailments combined. The maximum obtainable scores was 30 for the three non-condition specific items. Hence, the maximum obtainable score for the 12 competency elements for all the 14 minor ailments was 1290. In addition, the maximum obtainable score for each of the nine condition-specific competency element for the 14 minor ailments was 140.

## Data collection process

Introductory letters containing the title, purpose and the anticipated benefits of the study were sent to independent and chain pharmacies where community pharmacists were sampled. In addition, a short video focused on community pharmacists' management of minor ailment in UK was also provided to stimulate interest and enhance participation. The data collector provided the study participants with the self-administered questionnaires at their premises after signing the informed consent forms. Clarifications were provided when required and completed questionnaires were promptly collected. Reminders were sent via phone calls or text message to respondents who did not complete the questionnaires during the first visit.

## Ethics approval

The Qatar University's Institutional Review Board approved the study protocol before the commencement of data collection (QU-IRB reference number 1074-E/19, dated 03 May 2019).

## Data analysis

Data analysis was done with the SPSS version 26.0. for Windows (IBM SPSS Statistics for Windows, 2019, Version 26.0. Armonk, NY: IBM Corp). The demographic, workload and self-perceived competency data were tested for normality with the Shapiro-Wilk test (0.87, $p < 0.001$).

Frequencies, percentages, mean± SD or median (IQR) were used for descriptive statistics. Bivariate analysis of median self-perceived competency scores across gender and community pharmacy type was done with Mann-Whitney U test. Binary logistic regression was used to identify significant predictors of community pharmacists' self-perceived competency to manage minor ailments, and the significance level was set at ≤ 0.05.

## Results

Two hundred and eighty-two of the 307 study participants completed the questionnaire (response rate, 92.5%), and their demographic and workload characteristics are presented in Table 1. The majority of the respondents were males (68.1%; 192/282), within the age range of 31–40 years (55.3%; 156/282) and work for chains pharmacies (77.3%; 218/282). The median (IQR) duration of practice experience was 7 (4–10) and the most frequent pharmacy degree was BSc/BPharm (81.6%; 230/282). A majority of the community pharmacists (71.3%; 201/282) estimated that they attend to at least 30 customers per daily shift, while the most frequent estimate (54.6%; 154/282) of the number of customers with minor ailments per daily shift was 11–30. Consultation time of 6–10 minutes was the most frequent estimate (51.8%, 146/282) reported by community pharmacists for the management of minor ailments (Table 1).

Community pharmacists' self-perceived competency to manage the 14 selected common minor ailments in Qatar is as shown in Table 2. The minor ailments with the highest median competency score were: constipation (76), cold and catarrh (75), sore throat (74), headache (74), skin rash (74), and head lice. On the other hand, athlete foot (70), travel sickness (69) and ringworm (69) had the lowest median competency score.

The three condition-specific competency elements with the highest median competency score were recommendation of OTC medicines for the management of minor ailments (115), instructions to guide the use of recommended OTC medicines (115), and ability to define/describe minor ailments (115). Community pharmacists' ability to differentiate minor ailments from other similar medical conditions had the lowest self-perceived median competency score (104) (Table 3).

The median self-perceived competency score for the management of each of the 14 selected minor ailments was significantly higher in female community pharmacists relative to males, and among community pharmacists working for chain pharmacies relative to independent (p <0.05) (Table 4).

The median (IQR) for the total competency score (12 elements x 14 minor ailments) was 1031 (877–1123). The significant predictors of community pharmacists' self-perceived competency were female gender (OR = 2.39, 95%CI: 1.34–4.25, p = 0.003), and working for chain community pharmacies (OR = 2.54, 95%CI: 1.30–4.96, p = 0.006) (Table 5).

## Discussion

The minor ailments with the highest community pharmacists' self-perceived competency to manage were mainly constipation, cold and catarrh, headache and skin conditions; and these have been reported as some of the most commonly encountered minor ailments in both developed and developing countries including Qatar [4, 8, 9, 19, 29–31]. Furthermore, the specific competency elements with the highest median scores were the ability to recommend appropriate OTC medicines and provide instructions to guide its use; describe/define the 14 selected common minor ailments, and determine when referral to a physician is needed. In addition, competency elements such as the ability to identify the etiology, and signs and symptoms, recommend appropriate non-pharmacological measures, and recognize considerations for special populations had median scores that approximate 80% of the maximum obtainable score.

**Table 1. Community pharmacists' demographic and workload data related to minor ailments in Qatar (N = 282).**

| Item | n (%) |
|---|---|
| **Gender** | |
| Male | 192 (68.1) |
| Female | 90 (31.9) |
| **Age group (years)** | |
| 21–30 | 101 (35.8) |
| 31–40 | 156 (55.3) |
| 41–50 | 18 (6.4) |
| 51–60 | 5 (1,8) |
| >60 | 2 (0.7) |
| **Nationality** | |
| Indian | 122 (43.2) |
| Egyptian | 99 (35.1) |
| Sudanese | 25 (8.9) |
| Filipino | 21 (7.4) |
| Jordanian | 5 (1.8) |
| Syrian | 4 (1.4) |
| Pakistani | 4 (1.4) |
| Palestinian | 1 (0.4) |
| Canadian | 1 (0.4) |
| **Experience (years), Median (IQR)** | 7 (4, 10.3) |
| **Highest pharmacy degree** | |
| BSc/BPharm | 230 (81.6) |
| MSc Pharm | 29 (10.3) |
| PharmD | 10 (3.5) |
| Diploma | 13 (4.6) |
| **Type of community pharmacy** | |
| Independent | 64 (22.7) |
| Chains | 218 (77.3) |
| **# of Customers per daily shift** | |
| 1–10 Customers | 3 (1.1) |
| 11–20 Customers | 21 (7.4) |
| 21–30 Customers | 57 (20.2) |
| >30 Customers | 201 (71.3) |
| **# of customers with minor ailments per daily shift** | |
| 1–10 Customers | 69 (24.5) |
| 11–20 Customers | 80 (28.4) |
| 21–30 Customers | 74 (26.2) |
| >30 Customers | 59 (20.9) |
| **Consultation time for minor ailments** | |
| < = 5 minutes | 113 (40.1) |
| 6–10 minutes | 146 (51.8) |
| 11–15 minutes | 19 (6.7) |
| 16–20 minutes | 4 (1.4) |

These findings are probably due to combination of factors including experiences gathered in practice, exposure to undergraduate pharmacy curricular components including therapeutic and pharmaceutical care planning, and self-care of medical conditions that are minor and can

**Table 2. Community pharmacists' self-perceived competency to manage each of the common 14 minor ailments in Qatar (maximum obtainable score per minor ailment = 90).**

| Minor ailments | Median [IQR] |
|---|---|
| Constipation | 76 [65–82] |
| Cold and catarrh | 75 [67–82] |
| Headache | 74 [67–81] |
| Head lice | 74 [64–82] |
| Sore throat | 74 [65–81] |
| Skin rash | 74 [65–81.5] |
| Sun burn | 72 [61–81] |
| Teething discomfort | 72 [63–80] |
| Musculoskeletal pain | 72 [62–80] |
| Burns/scalds | 71 [61–79.5] |
| Hay fever | 71 [61–80] |
| Athlete's foot | 70 [59–78] |
| Ring worm | 69 [56–77] |
| Travel sickness | 69 [59–78] |

be managed effectively with the appropriate use of non-prescription medicines [32–34]. In addition, these components may also have been critical parts of the continuous professional development (CPD) programs for community pharmacists in Qatar and these may have contributed to the observed self-perceived competency levels [35, 36]. Hence, it is probably safe to conclude that community pharmacists are poised to assume the task of the effective management of common minor ailments in Qatar. This could potentially contribute to the reduction of clinical and financial burden associated with minor ailment-related hospital visits in Qatar, and enhance a more efficient use of available healthcare resources to achieve better value for money especially at the primary care and secondary care level.

Notwithstanding, a system that regularly audits community pharmacists' perceived competency to manage minor ailments is warranted, and this is because the current study showed that minor ailments such as travel sickness, ringworm and athlete foot that were identified as common in Qatar had relatively lower median self-perceived competency scores. In addition, the competency element with the lowest median score was the ability to differentiate minor ailments from other medical conditions with similar signs and symptoms. This finding appeared consistent with that of a recent study conducted in Qatar, which reported that some participants in an event diary using critical incident technique could not identify the differential

**Table 3. Community pharmacists' self-perceived competency for condition-specific elements (maximum obtainable score per competency element for 14 minor ailments = 140).**

| Competency element | Median [IQR] |
|---|---|
| Recommend appropriate OTC medicines | 117 [103–129] |
| Instruction on the use of recommended medicines | 115 [103–128] |
| Description / definition | 115 [102–126] |
| Determination of when referral is required | 114 [101–126] |
| Etiology | 112 [97.3–122.8] |
| Signs and symptoms | 112 [101–126] |
| Recommend appropriate non-pharmacological measures | 112 [97–125] |
| Recognize considerations for special populations | 112 [98–126] |
| Differentiate minor ailments from similar conditions | 109 [96–121] |

**Table 4. Comparison of community pharmacists' self-perceived competency to manage minor ailments across gender and community pharmacy types (N = 282].**

| Minor ailment | Male (n = 192) Median | Female (n = 90) Median | P-Value | Independent (n = 64) Median | Chain (n = 218) Median | P-Value |
|---|---|---|---|---|---|---|
| Constipation | 73.5 | 78 | 0.014* | 68 | 77 | 0.001* |
| Cold & catarrh | 73 | 78 | 0.017* | 71 | 76 | 0.003* |
| Hay fever | 70 | 74 | 0.023* | 64 | 72 | 0.001* |
| Headache | 72.5 | 77 | 0.029* | 71 | 76 | 0.001* |
| Teething discomfort | 71 | 76 | 0.001* | 64 | 73 | 0.001* |
| Musculoskeletal pain | 72 | 73 | 0.208 | 68 | 73 | 0.001* |
| Travel sickness | 68.5 | 71 | 0.126 | 61 | 71 | 0.001* |
| Head lice | 72. | 79 | 0.002* | 69 | 75 | 0.001* |
| Athlete's foot | 68.5 | 72 | 0.022* | 62 | 70.5 | 0.001* |
| Ring worm | 67 | 72 | 0.109 | 59 | 70 | 0.001* |
| Sore throat | 72 | 78 | 0.005* | 69 | 76 | 0.001* |
| Nappy rash | 72 | 78 | 0.003* | 70 | 75 | 0.001* |
| Burns /scalds | 69 | 74 | 0.003* | 63 | 72 | 0.001* |
| Sun burn | 70 | 76 | 0.002* | 64 | 74 | 0.001* |

Mann Whitney U test (p<0.05 (Significant).

**Table 5. Binary logistic regression of the predictors of community pharmacists' self-perceived competency to manage 14 selected common minor ailments in Qatar (N = 282).**

| Item | Categories (n) | Competency | | B | SE | Wald | Exp(B) | 95% CI for Exp(B) | | P-value |
|---|---|---|---|---|---|---|---|---|---|---|
| | | <median (<1032) n (%) | ≥median (≥1032) n (%) | | | | | Lower | Upper | |
| **Gender** | Male (192) | 110 | 82 | | | | 1 (reference) | | | |
| | Female (90) | 32 | 58 | 0.869 | 0.295 | 8.672 | 2.385 | 1.337 | 4.253 | 0.003* |
| **Age groups (years)** | ≤ 40 (257) | 132 | 125 | | | | 1 (reference) | | | |
| | > 40 (25) | 10 | 15 | 0.630 | 0.626 | 1.012 | 1.877 | 0.550 | 6.401 | 0.314 |
| **Nationality** | Arabs (134) | 72 | 62 | | | | 1 (reference) | | | |
| | Non-Arabs (148) | 70 | 78 | 0.199 | 0.279 | 0.510 | 1.220 | 0.706 | 2.108 | 0.475 |
| **Highest pharmacy degree** | BSc/BPharm (228) | 106 | 122 | | | | 1 (reference) | | | |
| | Non-BSc/BPharm (54) | 36 | 18 | -0.558 | 0.350 | 2.536 | 0.573 | 0.288 | 1.137 | 0.111 |
| **No. of customers per daily shift** | ≤ 30 (81) | 47 | 34 | | | | 1 (reference) | | | |
| | > 30(201) | 95 | 106 | 0.361 | 0.344 | 1.107 | 1.435 | 0.732 | 2.814 | 0.293 |
| **No. of customers with MAs per daily shift** | ≤ 20 (149) | 83 | 66 | | | | 1 (reference) | | | |
| | >20(133) | 59 | 74 | 0.237 | 0.306 | 0.601 | 1.267 | 0.696 | 2.307 | 0.438 |
| **Type of community pharmacy** | Independent (64) | 47 | 17 | | | | 1 (reference) | | | |
| | Chain (218) | 95 | 123 | 0.934 | 0.341 | 7.511 | 2.544 | 1.305 | 4.962 | 0.006* |

NB: 1032 (median score) is the cutoff point for self-perceived competency (maximum is 1290); MAs = Minor ailments B = Coefficient; SE = Standard Error; Exp(B) = Exponentiation of coefficient; CI = Confidence Interval

* p<0.05 (statistically significant).

diagnoses of minor ailments that may be out of community pharmacists' scope of practice and require referral to a physician [35]. Furthermore, the community pharmacists' self-perceived competency scores for a sizeable number of the competency elements approximates 80% of the maximum score obtainable. Hence, these suggest that the probability of actual competency gaps still exist and this information may be useful in the planning of future CPD programs focused on improving the capacity of community pharmacists in Qatar to manage minor ailment more effectively.

The significantly better median self-perceived competency scores among female community pharmacists relative to males, and the identification of the female gender as a significant determinant of community pharmacists' self-perceived competency to manage the 14 selected common minor ailments are being reported for the first time. The female gender had twice the odds of self-perceived higher competency score relative to male and this is despite that they constituted only about a third of the study participants. These findings are probably related to higher perceived self-efficacy among female community pharmacists as this is strongly related to self-perceived competency [20, 24]. In addition, female community pharmacists have been reported to have better clinical skills including the ability to communicate and establish good rapport, and gain patients' trust [37]. This may have contributed to their self-perceived competency to manage minor ailments more effectively.

Similarly, the significantly higher self-perceived competency scores observed among community pharmacists working in chain pharmacies, and the finding that working in chain pharmacies was also a significant determinant of community pharmacists' self-perceived competency to manage the 14 selected common minor ailments in Qatar is being reported for the first time. These findings are probably due to organizational policies and practice in chain pharmacies. This is because organizational policies and practice are significant predictors of employees' work-related behavior and job experience [38]. For instance, community pharmacists working in an organizational setting such as in chain pharmacies are more likely to have better access to training and development opportunities that may enhance their skills and on-the-job experience than those working in independent pharmacies. Hence, they are more like to have higher self-perceived competency to complete assigned tasks successfully, including those related to the management of minor ailments [20].

## Strengths and limitations

This is the first nationwide study that assessed community pharmacists' self-perceived competency to manage selected minor ailments commonly encountered in practice, and its significant predictors. The study findings may add to global knowledge in the study area and provide a basis for the development of community pharmacy-specific competency framework for providing minor ailments services especially in developing settings. The study has a few limitations including the use of non-probability sampling method. However, the purposive sampling method was based on actual proportional representation and this was chosen to mirror the sampling distribution of community pharmacists in Qatar. Furthermore, community pharmacists' response may have been affected by social desirability bias, as this was a self-administered survey. However, the high internal consistency of the questionnaire items probably suggests that the study findings are valid. In addition, social desirability bias appeared not to have affected the relatively lower self-perceived competency scores reported by community pharmacists for some of the selected minor ailments and competency elements. Lastly, another limitation is the use of numerical score to distill a complex construct such as cognitive skill, beliefs and attitude, which may or may not reflect the actual reality.

### Implication for policy and service planning

The current study has provided important insights into community pharmacists' self-perceived competency to manage minor ailments commonly encountered in practice, and its significant predictors, and these are crucial for the development of an appropriate institutional framework that will guide the ceding of the task of managing minor ailments to a primary care professional such as community pharmacists. For instance, the finding regarding the minor ailments with the highest self-perceived competency score by community pharmacists is potentially useful in identifying the initial list of minor ailments that will be appropriate for inclusion in the framework or scheme designed to guide the implementation of the policy of ceding the task of managing minor ailments to community pharmacists in Qatar. This is more likely to ensure that such a policy change is fit-for-purpose, meet societal needs and enhances effective service delivery. However, the structured institutional framework must include a referral mechanism with clearly defined criteria to promote patient safety and guide community pharmacists during the management of minor ailments. Furthermore, the gaps identified in community pharmacists' self-perceived competency is potentially useful in designing professional development programs focused on continuing improvement of the capacity of community pharmacists to manage minor ailment effectively and safely. Lastly, the insights provided for the first time by the current study about the significant predictors of community pharmacists' self-perceived competency is useful in identifying appropriate interventions that should be deployed to improve the readiness of community pharmacists to take over the task of managing minor ailments with the potential benefits of reducing the associated clinical and financial burden. For instance, assigning more prominent role to females community pharmacists and those working for chain pharmacies who seem to have higher self-perceived competency in designing interventions focused on improving community pharmacists' readiness to assume the role seems reasonable and may enhance effective service delivery.

## Conclusions

Community pharmacists' self-perceived competency appeared adequate for majority of the common minor ailments, and it was highest for the management of constipation, cold and catarrh, headache and skin conditions and specifically for the recommendation of OTC medicines and provision of instructions to guide its use. However, diagnostic ability to differentiate minor ailments from other medical conditions with similar features had the lowest median competency score. Female gender and working in chain pharmacies were the significant predictors of self-perceived competency to manage minor ailments.

## Author Contributions

**Conceptualization:** Ahmed Mohamed Makhlouf, Mohamed Izham Mohamed Ibrahim, Ahmed Awaisu, Saseendran Kattezhathu Vyas, Kazeem Babatunde Yusuff.

**Data curation:** Ahmed Mohamed Makhlouf, Mohamed Izham Mohamed Ibrahim, Kazeem Babatunde Yusuff.

**Formal analysis:** Ahmed Mohamed Makhlouf, Mohamed Izham Mohamed Ibrahim, Ahmed Awaisu, Saseendran Kattezhathu Vyas, Kazeem Babatunde Yusuff.

**Investigation:** Ahmed Mohamed Makhlouf, Mohamed Izham Mohamed Ibrahim, Kazeem Babatunde Yusuff.

**Methodology:** Ahmed Mohamed Makhlouf, Mohamed Izham Mohamed Ibrahim, Ahmed Awaisu, Saseendran Kattezhathu Vyas, Kazeem Babatunde Yusuff.

**Supervision:** Mohamed Izham Mohamed Ibrahim, Kazeem Babatunde Yusuff.

**Visualization:** Mohamed Izham Mohamed Ibrahim, Kazeem Babatunde Yusuff.

**Writing – original draft:** Ahmed Mohamed Makhlouf, Mohamed Izham Mohamed Ibrahim, Ahmed Awaisu, Saseendran Kattezhathu Vyas, Kazeem Babatunde Yusuff.

**Writing – review & editing:** Ahmed Mohamed Makhlouf, Mohamed Izham Mohamed Ibrahim, Ahmed Awaisu, Saseendran Kattezhathu Vyas, Kazeem Babatunde Yusuff.

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
