## [Decision Letter · Decision Letter 0]

30 May 2021

PONE-D-21-08313

Management of common minor ailments in Qatar: Community pharmacists’ self-perceived competency and its predictors

PLOS ONE

Dear Dr. Yusuff,

Thank you for submitting your manuscript to PLOS ONE. After careful consideration, we feel that it has merit but does not fully meet PLOS ONE’s publication criteria as it currently stands. Therefore, we invite you to submit a revised version of the manuscript that addresses the points raised during the review process.

We look forward to receiving your revised manuscript.

Kind regards,

Jenny Wilkinson, PhD

Academic Editor

PLOS ONE

Journal Requirements:

2. Please include additional information regarding the survey or questionnaire used in the study and ensure that you have provided sufficient details that others could replicate the analyses. For instance, if you developed a questionnaire as part of this study and it is not under a copyright more restrictive than CC-BY, please include a copy, in both the original language and English, as Supporting Information. Moreover, please include more details on how the questionnaire was pre-tested, and whether it was validated.

3. Please correct your reference to "p=0.000" to "p<0.001" or as similarly appropriate, as p values cannot equal zero.

Additional Editor Comments:

Thank you for your submission. The reviewers have provided comments and suggests to strengthen your work and provide readers with a better understanding of both the methods used and the interpretation and context of your findings.

Reviewers' comments:

Reviewer's Responses to Questions

**Comments to the Author**

1. Is the manuscript technically sound, and do the data support the conclusions?

Reviewer #1: Partly

Reviewer #2: Yes

2. Has the statistical analysis been performed appropriately and rigorously? 

Reviewer #1: I Don't Know

Reviewer #2: I Don't Know

3. Have the authors made all data underlying the findings in their manuscript fully available?

Reviewer #1: Yes

Reviewer #2: Yes

4. Is the manuscript presented in an intelligible fashion and written in standard English?

Reviewer #1: Yes

Reviewer #2: Yes

5. Review Comments to the Author

Reviewer #1: 1) Some grammatical mistakes, like Line 85, where is should read "The literature search ...". Others like this were evident throughout, which I did not focus on here.

The big issues will be first ...

2) The conglomeration of numbers to arrive at 1260 and 1290 can be figured out by readers, but I don't think I would call it intuitive. Also, the only mention of this global score was at Line 254, however one can assume the numbers in Table 3 would accrue up to that value. Either way, given the attention it got in the Methods, it got very little attention in the Results. That makes understanding Table 3 pretty tricky.

3) In Lines 167-168, which items of the "final questionnaire" were included in the Cronbach's alpha measure? If it included such things as the Demographics, then we have a problem (those items would not be "internally consistent" with other items). IF the 12 items of competence were the source, then there is some hope. That said, however, you have 12 quite diverse items in that list, and the question is -- are they truly all ONE construct that can actually have internal consistency? Either way, later in Line 345, you simply CANNOT say you attained a "valid and reliable" survey based on that score alone. Even with the input from 3 faculty and 1 practitioner for content validity, you have to tone down your choice of words here. Regarding reliability, authors often confuse "reliability" of Cronbach's alpha with actual test-retest reliability, where a second measure is taken to see if its reliable over time.

4) Not enough attention is devoted to the 1-10 Likert Scale. All we know are the poles -- LOW and FULL. What about the other 8 points on the scale? Did you use any verbal descriptors?

5) A critical limitation of all survey research is trying to distill complex attitudes/beliefs down to numbers (as in, a scale of some sort). Using a scale to measure competence has to be recognized as a Limitation.

6) On Line 337, you claim you have 'added significantly' to the field. Its good to be optimistic, but we often don't reach that level. I would be more introspect.

7) I think the editors should have a statistician check out Table 5, to make sure all is on solid ground.

8) Related to the 1-10 scale, you went with them being ordinal rather than interval. Many do that. Many others go with intervality, thus jumping to means. That explains the use of Medians for your Results. Just a comment, no changes suggested. However, I would wonder whether mean years of practice would be better than IQR (Line 207).

9) For Lines 208-211, with at least 30 encounters occurring per shift, I was surprised to see 11-30 being the number for minor ailments. That means most in a day were of this sort. Just an observation.

10) Regarding how the list of Minor Ailments was created, I can live with the references cited (17-19,27-31), except that refs 19 and 31 seem like outliers here (just one condition each -- VGE and headache). In your approach, you could have taken to how others have defined Minor Ailments, which then leads to a list of possible candidates (as would have been described in ref 27).

As an aside, I found it odd that acne, warts, diarrhea, cold sores, heartburn did not surface. But, you get what you get. Of note, Athlete Foot should read Athlete's Foot.

11) For the sample size calculation, I agree with the method. However, one should note that the "response distribution of 50%" literally refers to surveys where only a FOR/AGAINST option is available. That is, a 50/50 proposition for a response. In your case, you would not have such a set-up. That stated, no changes suggested here.

12) For Lines 169-176, it is tricky to follow the 12 competency elements -- 9 condition and 3 non-condition. Table 3 lists out the 9, fine. At Lines 159-161, the non-condition ones are listed. Thereafter for those 3, only special populations appears (Line 281). The others are basically ignored. Furthermore, I think you have 4 of those, not 3 -- Documentation and Use of Resources are clearly separate things.

13) I am not sure you found all the key reports to cite: a) Stewart IJPP 2009, 17: 89-94, b) George Ann Pharmaco 2006, 40: 1843-50, c) Hoti Pharm World Sci 2010, 32: 610-21, d) Taylor SelfCare 2016, 7(1): 10-21 ... might be of some use.

14) In Table 1, the Experience line, you have "(years)", "(IQR)", but also "(%)" at the top of the column, making it hard to understand the listing of "7 (4, 10.3)".

15) In Table 3, I think you have some unintended consequences. "Description/Definition" could easily overlap with "Signs and Symptoms".

16) Lines 313-314 -- I don't think you want to qualify the data here with use of "notwithstanding". The lower numbers of women is not an issue here, you simply mention the discrepancy seen, which stands on its own.

Reviewer #2: Many thanks for the opportunity to review this manuscript investigating the self-efficacy or self-perception of competency in managing minor ailments in pharmacies in Qatar. I am not convinced of the need for and the contribution of this work. I describe some observations below to illustrate this concern.

From the outset of the manuscript, the size or real extent of the problem that the findings of this study aim to address is not entirely convincing. Is it a case in Qatar that minor ailments are being referred inappropriately out of pharmacy and therefore contributing the burden in ED and GPs? In the discussion, it is presented that self-efficacy was generally high and authors comment that this has the potential to relieve the burden from elsewhere in the system. I would contest that based on the core training of pharmacists globally, there is an acknowledgement and expectation that pharmacists can manage minor ailments, so this study has not really addressed or answered a pressing issue or question. In fact it is in most cases, the structure of the healthcare systems, health-seeking behaviours of the population and resource accessibility in community pharmacy that lead to overburdened ED and primary care physicians being consulted for minor ailments.

Pharmacists are not conventionally diagnosticians (therefore unsurprising that ability to differentiate minor ailment from other conditions was the lowest rated competency score), therefore the management of minor ailments has always been on the premise of management of symptoms and escalation/referral where 'red flags' or problems fall outside of the competence of a pharmacist. Therefore, when minor ailments are referred to, there needs to be a description about the nature and acuity of that ailment. Asking a pharmacist whether they can manage constipation is ambiguous if there are no other details about the patient and the ailment, e.g. how long experienced the symptoms, recent travel abroad, with other symptoms, etc. So if a pharmacist rates themselves as less confident/competent, are they really rating themselves on what the researcher thinks or are they thinking about the unknowns surrounding that condition?

In brief, what does the quantitative analysis of pharmacist's self-efficacy really tell us? They are risk averse? They are not comfortable with the ambiguity in the questions? There are deficiencies in the training of that pharmacist?

The authors suggest that there is a competency gap due to the lower score on the ability to undertake differential diagnoses. I think this assumption does not acknowledge the generally low access of community pharmacists to patient clinical records that would facilitate clinical and therapeutic reasoning. So is it all about competency?

The implications for policy and practice are overly simplified. The understanding of self-perceived competency is not sufficient information on which to build a national strategy/framework for managing minor ailments. There needs to be acknowledgement that minor ailments exist on a continuum of acuity, and it is about understanding what community pharmacists can do to contribute to the overall load. Shifting all patients with constipation to community pharmacy may not be the most safe and appropriate recommendation simply based on these findings. Which types of patients? What grades of constipation? What are the thresholds for referrals? These are going to be really important to build a framework.

It is worth reflecting on whether self-efficacy equates to safer, more effective practice and outcomes. Because females are more confident, is the assumption they are best placed to manage minor ailments in community pharmacy? I think this needs some more consideration. Similarly, it would be interesting to investigate why those in chain pharmacies feel more confident, and if their performance in managing minor ailments is indeed better. So the findings of this work offer further areas to investigate before being very helpful in informing the deign of an intervention/framework.

Some other more specific comments:

The first few lines of the abstract are overly convoluted and could be expressed much simpler. What is the problem that this study aims to address/investigate?

Introduction

Lines 87-98 I appreciate that the authors have linked the aim of this study to Bandura's SCT in order to justify measuring self-competency. However, the link and significance to practice is not entirely convincing.

Was content validity really assessed? If so, how was this done? And the results need to be included. Or was it face validity?

6. PLOS authors have the option to publish the peer review history of their article (what does this mean?). If published, this will include your full peer review and any attached files.

Reviewer #1: No

Reviewer #2: No

---

## [Author Response · Author response to Decision Letter 0]

8 Jun 2021

Response to Review Comments

05 June 2021

The Editor-In-Chief

PLOS ONE

Dear Sir,

Re: Manuscript ID PONE-D-21-08313 – “Management of common minor ailments in Qatar: Community pharmacists’ self-perceived competency and its predictors”

Our sincere thanks for the opportunity to revise the manuscript ID PONE-D-21-08313 – “Management of common minor ailments in Qatar: Community pharmacists’ self-perceived competency and its predictors” which is under your consideration for publication in PLOS ONE. We thank the reviewers and the editor for the insightful comments and useful suggestions and we have revised the manuscript accordingly. Please find stated below our point-by-point response to all the comments.

EDITOR’S COMMENTS

2. Please include additional information regarding the survey or questionnaire used in the study and ensure that you have provided sufficient details that others could replicate the analyses. For instance, if you developed a questionnaire as part of this study and it is not under a copyright more restrictive than CC-BY, please include a copy, in both the original language and English, as Supporting Information. Moreover, please include more details on how the questionnaire was pre-tested, and whether it was validated.

3. Please correct your reference to "p=0.000" to "p<0.001" or as similarly appropriate, as p values cannot equal zero.

Response:

1. We have ensured that our manuscript meets all PLOS ONE’s style requirements. 

2. We have added additional information about the procedure used for the development and validation of the survey questionnaire. The questionnaire is not under a copyright and it is available upon reasonable request. This information was added to the “questionnaire development and structure” sub-section of the “Methods” section.

3. The recommended correction of the reference to p values has been done in Table 4 [line 253].

ADDITIONAL EDITOR COMMENTS:

Thank you for your submission. The reviewers have provided comments and suggests to strengthen your work and provide readers with a better understanding of both the methods used and the interpretation and context of your findings.

Response: The sections of the manuscript related to the methods, interpretation and context of 

the results presented have been revised to strengthen the work and enhance readers’ understanding as suggested [See response to reviewers’ comments]. 

REVIEWERS’ COMMENTS

Reviewer #1:

Comment-1) Some grammatical mistakes, like Line 85, where is should read "The literature search ...". Others like this were evident throughout, which I did not focus on here.

Response-1: Sincere gratitude to the reviewer for the identification of the grammatical mistakes as correcting these will improve the clarity and readability of the manuscript. The entire manuscript has been revised to identify and correct these mistakes.

Comment-2) The conglomeration of numbers to arrive at 1260 and 1290 can be figured out by readers, but I don't think I would call it intuitive. Also, the only mention of this global score was at Line 254, however one can assume the numbers in Table 3 would accrue up to that value. Either way, given the attention it got in the Methods, it got very little attention in the Results. That makes understanding Table 3 pretty tricky.

Response 2: Sincere thanks to the reviewer for the comment. The details regarding how the maximum obtainable scores for the 9 condition-specific and 3 non-condition specific elements were calculated were adequately presented in the method section [line 174-181] as alluded to by the reviewer. In addition, the relevant details regarding the median (IQR) for the total competency score (12 elements x 14 minor ailments) was presented in line 256 - 257. This was done to ensure clarity and ease of understanding of Tables 2, 3 and 5.

Comment-3) In Lines 167-168, which items of the "final questionnaire" were included in the Cronbach's alpha measure? If it included such things as the Demographics, then we have a problem (those items would not be "internally consistent" with other items). IF the 12 items of competence were the source, then there is some hope. That said, however, you have 12 quite diverse items in that list, and the question is -- are they truly all ONE construct that can actually have internal consistency? Either way, later in Line 345, you simply CANNOT say you attained a "valid and reliable" survey based on that score alone. Even with the input from 3 faculty and 1 practitioner for content validity, you have to tone down your choice of words here. Regarding reliability, authors often confuse "reliability" of Cronbach's alpha with actual test-retest reliability, where a second measure is taken to see if its reliable over time.

Response-3: Heartfelt thanks to the reviewer for this excellent observation. Demographics were not included in the Cronbach alpha assessment. Only the 12 items used for the assessment of self-perceived competency were included. The manuscript has been revised to clarify this [line 167 -169]. We thank the reviewer for the comment regarding the diverse but comprehensive nature of the 12 items used for the assessment of self-perceived competency. We assert with all due respect that all the items are appropriate for the global assessment of the competency required by community pharmacists to manage minor ailments effectively. We acknowledge the reviewer’s concern with our use of the phrase “valid and reliable” in line 345. We were cognizant of not overstating the reliability of the questionnaire used in our study and this was why we have added the phrase “probably suggest” to tone down our choice of words as suggested by the reviewer [line 348]. However, Cronbach alpha is a measure of internal consistency, which is one of the methods used extensively for reliability analysis in survey research. We agree with the reviewer that test-retest is another option that can be used, but we settled for Cronbach alpha, as it is also a valid measure of reliability analysis in survey research. 

Comment-4) Not enough attention is devoted to the 1-10 Likert Scale. All we know are the poles -- LOW and FULL. What about the other 8 points on the scale? Did you use any verbal descriptors?

Response-4: Many thanks to the reviewer. Detailed instructions were provided with sufficient clarity to respondents in the questionnaire regarding the rating of self-perceived competency on the 10-point scale (1: low competence, 10: full competence). There were no ambiguity and no respondent expressed any difficulty with using the scale. 

Comment-5) A critical limitation of all survey research is trying to distill complex attitudes/beliefs down to numbers (as in, a scale of some sort). Using a scale to measure competence has to be recognized as a Limitation.

Response-5: We thank the reviewer and concur with this observation. We respectfully submit that these issues were addressed in the limitation section [343 – 350]. However, the suggested limitation has been added to the revised manuscript [Line 350-351].

Comment-6) On Line 337, you claim you have 'added significantly' to the field. Its good to be optimistic, but we often don't reach that level. I would be more introspect.

Response-6: We totally agree with the need for circumspection and this was why we have revised the phrase as “may add significantly to global knowledge” [line 339].

Comment-7) I think the editors should have a statistician check out Table 5, to make sure all is on solid ground.

Response-7: Sincere thanks to the reviewer for this comment. We agree with this and the reviewer can be rest assured that the logistic regression analysis was adequately done.

Comment-8) Related to the 1-10 scale, you went with them being ordinal rather than interval. Many do that. Many others go with intervality, thus jumping to means. That explains the use of Medians for your Results. Just a comment, no changes suggested. However, I would wonder whether mean years of practice would be better than IQR (Line 207).

Response-8: Heartfelt thanks to the reviewer for this excellent observation. However, median [IQR] was used for the years of practice because test of data normality with Shapiro Wilk test showed non-normal distribution. This was stated in the manuscript [line 199 -201].

Comment-9) For Lines 208-211, with at least 30 encounters occurring per shift, I was surprised to see 11-30 being the number for minor ailments. That means most in a day were of this sort. Just an observation.

Response-9: Yes, we concur with the reviewer on this point and were also thrilled by this finding.

Comment-10) Regarding how the list of Minor Ailments was created, I can live with the references cited (17-19,27-31), except that refs 19 and 31 seem like outliers here (just one condition each -- VGE and headache). In your approach, you could have taken to how others have defined Minor Ailments, which then leads to a list of possible candidates (as would have been described in ref 27).

Response-10: We thanks the reviewer for this observation. References 19 and 31 were just two of the eight references used for developing the initial list of minor ailments used in the first phase of the study, and they were included because they were conducted in similar study settings. The other references including ref 27 were also used. We are confident that the approach used for the selection of the final list of 14 minor ailments is appropriate.

Comment-10b) As an aside, I found it odd that acne, warts, diarrhea, cold sores, heartburn did not surface. But, you get what you get. Of note, Athlete Foot should read Athlete's Foot.

Response-10b: Many thanks to the comment and this is perfectly understandable. We were mainly concerned with reporting what we found, and this was what we did. The correction regarding athlete’s foot has been done [line 234, line 253].

Comment-11) For the sample size calculation, I agree with the method. However, one should note that the "response distribution of 50%" literally refers to surveys where only a FOR/AGAINST option is available. That is, a 50/50 proposition for a response. In your case, you would not have such a set-up. That stated, no changes suggested here.

Response-11: Heartfelt thanks to the reviewer. A conservative estimate of 50% is often used to ensure the calculation of the largest sample size required to conduct an adequately powered survey.

Comment-12) For Lines 169-176, it is tricky to follow the 12 competency elements -- 9 condition and 3 non-condition. Table 3 lists out the 9, fine. At Lines 159-161, the non-condition ones are listed. Thereafter for those 3, only special populations appears (Line 281). The others are basically ignored. Furthermore, I think you have 4 of those, not 3 -- Documentation and Use of Resources are clearly separate things.

Response-12: Sincere thanks to the reviewer for this excellent observation. We agree with the reviewer that all the elements were listed in 156-163. However, we noted an error in the listing of the items. The manuscript has been revised to correct the error [line 161-162]. The items listed in line 281 are not non-condition elements. Recognition of consideration for special population is part of the condition-specific elements. This correction has been done in the revised manuscript [line 161].

Comment-13) I am not sure you found all the key reports to cite: a) Stewart IJPP 2009, 17: 89-94, b) George Ann Pharmaco 2006, 40: 1843-50, c) Hoti Pharm World Sci 2010, 32: 610-21, d) Taylor SelfCare 2016, 7(1): 10-21 ... might be of some use.

Response-13: Heartfelt thanks to the reviewer for this kind suggestion. We noted that the focus of three of the suggested studies are not similar to ours. For instance, George Ann Pharmaco 2006, Hoti Pharm World Sci 2010 and Stewart IJPP 2009 were essentially focused on supplemental prescribing by pharmacists, including for more serious medical conditions. Lastly, an article with a similar focus, which was authored by Taylor JG, has already been cited in the manuscript [line 438]. 

Comment-14) In Table 1, the Experience line, you have "(years)", "(IQR)", but also "(%)" at the top of the column, making it hard to understand the listing of "7 (4, 10.3)".

Response-14: We thank the reviewer for this observation. Experience was the only outlier that was not a frequency count on Table 1 and we denoted this by adding “Median (IQR)” to that row.

Comment-15) In Table 3, I think you have some unintended consequences. "Description/Definition" could easily overlap with "Signs and Symptoms".

Response-15: Many thanks to the reviewer for the observation. We acknowledge that both items are related, but we are certain that the description / definition of a minor ailment is quite distinct from its signs and symptoms. 

Comment-16) Lines 313-314 -- I don't think you want to qualify the data here with use of "notwithstanding". The lower numbers of women is not an issue here, you simply mention the discrepancy seen, which stands on its own.

Response-16: We concur with the reviewer’s observation and the word “notwithstanding” has been replaced [line 315]. 

Reviewer #2: 

Comment-1) Many thanks for the opportunity to review this manuscript investigating the self-efficacy or self-perception of competency in managing minor ailments in pharmacies in Qatar. I am not convinced of the need for and the contribution of this work. I describe some observations below to illustrate this concern.

From the outset of the manuscript, the size or real extent of the problem that the findings of this study aim to address is not entirely convincing. Is it a case in Qatar that minor ailments are being referred inappropriately out of pharmacy and therefore contributing the burden in ED and GPs? In the discussion, it is presented that self-efficacy was generally high and authors comment that this has the potential to relieve the burden from elsewhere in the system. I would contest that based on the core training of pharmacists globally, there is an acknowledgement and expectation that pharmacists can manage minor ailments, so this study has not really addressed or answered a pressing issue or question. In fact it is in most cases, the structure of the healthcare systems, health-seeking behaviours of the population and resource accessibility in community pharmacy that lead to overburdened ED and primary care physicians being consulted for minor ailments.

Response-1: We thank the reviewer for this comment. However, we respectfully submit that the details regarding the study’s aim and justification were clearly articulated in the introduction section of the manuscript [Line 67-75, 77-85, 87-92]. Our extensive search of the literature showed that the use of collaborative models of practice which cede the task of managing minor ailments within a structured framework are available in developed settings, but rare in developing countries including Qatar. However, Qatar is one of the countries which is currently implementing a National Health Strategy (QNHS 2018-2022) focused on the provision of a functional patient-centered primary care service; and community pharmacists have been identified as a key healthcare professionals whose active participation is crucial to expanding access to functional primary care services including the effective management of minor ailments. However, studies focused on the comprehensive assessment of the self-perceived competency of community pharmacists to manage minor ailments are scanty. This was considered an essential first step as community pharmacists’ ability to assume the responsibility of managing minor ailments within a structured framework is substantially dependent on their self-perceived competence to execute the tasks associated with that responsibility successfully.

Comment-2) Pharmacists are not conventionally diagnosticians (therefore unsurprising that ability to differentiate minor ailment from other conditions was the lowest rated competency score), therefore the management of minor ailments has always been on the premise of management of symptoms and escalation/referral where 'red flags' or problems fall outside of the competence of a pharmacist. Therefore, when minor ailments are referred to, there needs to be a description about the nature and acuity of that ailment. Asking a pharmacist whether they can manage constipation is ambiguous if there are no other details about the patient and the ailment, e.g. how long experienced the symptoms, recent travel abroad, with other symptoms, etc. So if a pharmacist rates themselves as less confident/competent, are they really rating themselves on what the researcher thinks or are they thinking about the unknowns surrounding that condition?

In brief, what does the quantitative analysis of pharmacist's self-efficacy really tell us? They are risk averse? They are not comfortable with the ambiguity in the questions? There are deficiencies in the training of that pharmacist?

The authors suggest that there is a competency gap due to the lower score on the ability to undertake differential diagnoses. I think this assumption does not acknowledge the generally low access of community pharmacists to patient clinical records that would facilitate clinical and therapeutic reasoning. So is it all about competency?

Response-2: We thank the reviewer for the valuable observation stated above. However. We respectfully assert that the undergraduate training of pharmacists equip them with the diagnostic competence to identify and manage minor ailments. Two key components of these competencies include information gathering and counseling practices, which are within the core direct patient care competencies of pharmacists and should enable them to identify, assess and manage minor ailments effectively. Hence, the notion about community pharmacists’ lack of access to patients’ clinical records does not really apply in the management of minor ailments. This is more related to more serious and often chronic medical conditions requiring probably long term and often complicated drug therapy. As the reviewer alluded, whenever a pharmacist notes red-flags, they have always been taught to move the next step up the ladder, which is referral to the appropriate healthcare provider. In addition, we will like to emphasize that the respondents were not just simply asked whether they can manage a minor ailment such constipation or not. The details regarding the 12 condition- and non-condition specific competency elements used for the assessment were clearly stated in the manuscript [line 156-163].

Comment-3) The implications for policy and practice are overly simplified. The understanding of self-perceived competency is not sufficient information on which to build a national strategy/framework for managing minor ailments. There needs to be acknowledgement that minor ailments exist on a continuum of acuity, and it is about understanding what community pharmacists can do to contribute to the overall load. Shifting all patients with constipation to community pharmacy may not be the most safe and appropriate recommendation simply based on these findings. Which types of patients? What grades of constipation? What are the thresholds for referrals? These are going to be really important to build a framework.

Response-3: We are sincerely grateful for the valuable insights provided by the reviewer. We respectfully submit that our brief discussion of the implications of the study findings and recommendations were anything but oversimplified. We were specifically focused on explaining how the key findings including the common minor ailments identified in Qatar, the minor ailments with the highest self-perceived competency rating by community pharmacists, and the gaps identified in the competency elements. In addition, we also briefly discussed how these findings could be used to design an institutional framework for guiding the ceding of the task for managing minor ailments to community pharmacists in Qatar. In addition, we also recommended, based on the logistic regression analysis, giving prominent roles to the significant predictors of community pharmacists’ self-perceived competency in designing interventions focused on improving the management of minor ailment by community pharmacists in Qatar. However, we concur with the reviewer’s comment regarding the necessity to ensure that the structured framework will clearly spell out the criteria for referral by community pharmacists during the management of minor ailments. This has been added to the manuscript [line 363-366].

“However, the structured institutional framework must include a referral mechanism with clearly defined criteria to promote patient safety and guide community pharmacists during the management of minor ailments.”

Comment-4) It is worth reflecting on whether self-efficacy equates to safer, more effective practice and outcomes. Because females are more confident, is the assumption they are best placed to manage minor ailments in community pharmacy? I think this needs some more consideration. Similarly, it would be interesting to investigate why those in chain pharmacies feel more confident, and if their performance in managing minor ailments is indeed better. So the findings of this work offer further areas to investigate before being very helpful in informing the design of an intervention/framework.

Response 4: We thank the reviewer for this observation and we concur that our study has thrown up more leads for further research. However, we respectfully submit that we did not assert anywhere in the manuscript, that self-efficacy EQUATES safer and more effective practices and outcomes. Of course, other human and/or system-related factors will come into play. However, we state respectfully that self-efficacy is a valid and reliable determinant of task performance as espoused in Albert Bandura’s social cognitive theory; and this connection is well documented in published literature as we articulated in the manuscript [line 92-100]. Therefore, an assessment of community pharmacists’ self-perceived competency is appropriate and this is an essential first step in determining if an individual will be able to deliver effectively on a task.

Comment-5) Some other more specific comments:

The first few lines of the abstract are overly convoluted and could be expressed much simpler. What is the problem that this study aims to address/investigate?

Response-5: Heartfelt thanks to the reviewer. The abstract has been revised for clarity in accordance with the reviewer’s suggestion [line 31-35].

Comment-6) Introduction

Lines 87-98 I appreciate that the authors have linked the aim of this study to Bandura's SCT in order to justify measuring self-competency. However, the link and significance to practice is not entirely convincing.

Response-6: Many thanks to the reviewer. This comment has been addressed in our response to comment-4 above. 

Comment-7) Was content validity really assessed? If so, how was this done? And the results need to be included. Or was it face validity?

Response-7: Many thanks to the reviewer. Content validity of the survey tool was assessed as stated in the manuscript in line 165-169. However, the manuscript has been revised to enhance the clarity of the details of the procedure used [line 165-169]. Face validity is considered a component of content validity assessment, which is however regarded as too informal and relatively less objective. 

Reviewer #3: 

Comment -1) Introduction section: Majority of the studies cited and highlighted in the introduction section on minor ailments were from overseas and lacking for the local study. The explanation on local study only highlighted the general strategic goals in the area. It was also unclear why Bandura Theory was selected instead of other theories. The objectives of the current study were to conduct a baseline assessment of the self-perceived competency of community pharmacists to manage selected minor ailments. It was also unclear the purpose of conducting the baseline assessment.

Response-1: Sincere thanks to reviewer for this observation. However, we state with all due respect that contrary to the reviewer’s comment, a local study that was directly related to our study objective was cited. In addition, other studies that were conducted in settings that are similar to Qatar were also cited in the Introduction section of the manuscript [line 87-88], and these studies had nothing to do with the strategic goals of the QNV or QNHS. Citations of studies done overseas [developed settings] were correctly made regarding the use of a collaborative model of practice that involved the ceding of the task of managing minor ailments to community pharmacists within a structured framework. This is appropriate as no such published studies from developing settings including Qatar currently exist. The justification for the choice of Bandura social cognitive theory was clearly explained in line 89-100. We state with all due respect that this is an appropriate theoretical framework for our study. However, we agree with the reviewer that there may other theories that could be used but there nothing wrong with our choice of Albert Bandura and we think it suffices.

Comment-2) Method section: The first phase of the study is confusing. A standard reference should be used in identifying minor ailments. Only then the list can be amended based on the local study. In the first phase of the study it was highlighted that the development of the common minor ailments only involved physicians. Since the study will involve pharmacists at the community setting, their feedback is important to obtain. It was unclear why only physicians were included in the first phase of the study. Since the physicians were selected from the hospital settings they may rank the most commonly found minor ailments in the hospital rather than in the community pharmacy settings. Since there is no representative from the community pharmacists, there is a major concern in this area. In the second phase of the study it was unclear how the survey was distributed either via online or postage.

Response-2: Many thanks to the reviewer for this comment. A thorough review of relevant literature was done to identify the 8 standard references used for the development of the initial list of 58 minor ailments that were used for the first phase [line 128-129; line 149-150]. Furthermore, we believe that the details regarding the first phase of the study was made with sufficient clarity in line 121-134. Physicians that were sampled from healthcare settings where high clinical burden due to minor ailments are well documented and sampled community pharmacists were involved in the assessment of the suitability of common minor ailments that were selected for the second phase of the study [line 128-134]. The details of how the questionnaires were distributed were clearly stated in line 187-191].

Comment-3) Result section: Refer to table 3: Usually the competency element should include a statement with verbs such as able to recommend therapy. It is unclear with some of the competency elements in table 3 e.g etiology, signs and symptoms what are the competencies that need to be achieved.

Response-3: Sincere thanks to the reviewer. The action verb “I am competent to ..” was used in the questionnaire. The respondents ranked their self-perceived competency on scale of 1 – 10 (1: low competence, 10: full competence) for all the 12 condition and non-condition specific competency elements. This was clearly stated in line 174-175.

Comment-4) Discussion section: It is unclear on the statement highlighted from line 367-371: 'For instance, assigning more prominent role to females community pharmacists and those working for chain pharmacies who seem to have higher self-perceived competency in designing interventions focused on improving community pharmacists’ readiness to assume the role seems reasonable and may enhance effective service delivery.' Does self-perceived competency reflect the 'true' competencies of the person and how this has led the author to conclude such recommendation.

Response-4: Many thanks for the reviewer’s comment regarding line 367-371. Our recommendation was appropriately made based on the logistic regression analysis which identified these two factors as significant predictors of community pharmacists’ self-perceived competency to management minor ailments. We agree with the reviewer’s comment that self-perceived competency may or may not reflect actual practice and we did not make such a claim in the manuscript. In fact, this was why we chose the phrase “may enhance” in line 374 as we are cognizant that other factors may confound this relationship. However, it is well documented that self-perceived competency is valid determinant of task performance [line 92-100].

Comment-5) Conclusion section: It is unclear on the conclusion of the study findings: Example: 'Community pharmacists’ self-perceived competency appeared adequate for majority of the common minor ailments, and it was highest for the management of constipation, cold and catarrh, headache and skin conditions and specifically for the recommendation of OTC medicines and provision of instructions to guide its use.' In competency statement, the usual term used is either the person is competent or incompetent. So when the term used is adequate competency this can be confusing to the reader.

Response-5: Many thanks to the reviewer the comment. However, our conclusion is valid for and consistent with the key findings reported [line 223-226; 236-241; 257-259]. We respectfully differ from the reviewer’s assertion that competency assessment is based on a all or none approach [either competent or incompetent]. Competency, especially in the clinical realm, often involve the use of a combination of skill set required to effectively complete a specific task. Hence, a person may have adequate competency for certain skill set but not the other. This is similar to the findings of our study as stated in line 223-226 [Table 2] and 236-241[Table 3].

---

## [Decision Letter · Decision Letter 1]

26 Jul 2021

PONE-D-21-08313R1

Management of common minor ailments in Qatar: Community pharmacists’ self-perceived competency and its predictors

PLOS ONE

Dear Dr. Yusuff,

Thank you for submitting your manuscript to PLOS ONE. After careful consideration, we feel that it has merit but does not fully meet PLOS ONE’s publication criteria as it currently stands. Therefore, we invite you to submit a revised version of the manuscript that addresses the points raised during the review process.

We look forward to receiving your revised manuscript.

Kind regards,

Jenny Wilkinson, PhD

Academic Editor

PLOS ONE

Journal Requirements:

Additional Editor Comments:

Thank you for your revisions. Some additional comments are provided for your consideration.

Reviewers' comments:

Reviewer's Responses to Questions

**Comments to the Author**

1. If the authors have adequately addressed your comments raised in a previous round of review and you feel that this manuscript is now acceptable for publication, you may indicate that here to bypass the “Comments to the Author” section, enter your conflict of interest statement in the “Confidential to Editor” section, and submit your "Accept" recommendation.

Reviewer #1: (No Response)

2. Is the manuscript technically sound, and do the data support the conclusions?

Reviewer #1: Partly

3. Has the statistical analysis been performed appropriately and rigorously? 

Reviewer #1: Yes

4. Have the authors made all data underlying the findings in their manuscript fully available?

Reviewer #1: Yes

5. Is the manuscript presented in an intelligible fashion and written in standard English?

Reviewer #1: Yes

6. Review Comments to the Author

Reviewer #1: I am fine with the corrections made, except for the following:

I would advise against the approach you took in your rebuttal to reviewer comments. On several instances, the feeling appeared to be 'while we thank the reviewer for the comment, we did it the right way' and will disregard it.

For example, regarding the Likert scale sequence, it will not suffice to say that responders were 'given detailed instructions' and things were successful. Likert scales are very tricky devices, and to ask for input into what came b/n Low and Full was a reasonable request to make.

Secondly, to state that Description and Signs/Symptoms were sufficiently distinct items, I would love to be enlightened. When I think of the Description of the Common Cold, I conjure up "a viral upper respiratory infection manifesting as nasal congestion, runny nose, perhaps a cough, maybe a headache etc. Now, how do signs/symptoms deviate from that?

For line 346, the statement -- "probably suggests the study findings are valid and reliable" -- based on a sole Cronbach's alpha is too far-reaching. I like your items in Table 3. Regardless of that stance, do you really feel you got things so right in the survey to make that conclusion?? I look back to all the OTC research I have done over 30 years and I STILL worry about what mistakes I made.

There are many others, but I will let it rest.

Lastly, to downgrade "significantly add to the literature" to "may significantly add" is still clearly over-valuing the work you created. It is just not that profound, similar to the rest of us working in this area.

7. PLOS authors have the option to publish the peer review history of their article (what does this mean?). If published, this will include your full peer review and any attached files.

Reviewer #1: No

---

## [Author Response · Author response to Decision Letter 1]

27 Jul 2021

Response to Review Comments

27 July 2021

The Editor-In-Chief

PLOS ONE

Dear Sir,

Re: Manuscript ID PONE-D-21-08313R1 – “Management of common minor ailments in Qatar: Community pharmacists’ self-perceived competency and its predictors”

Our sincere thanks for the opportunity to revise the manuscript ID PONE-D-21-08313R1 – “Management of common minor ailments in Qatar: Community pharmacists’ self-perceived competency and its predictors” which is under your consideration for publication in PLOS ONE. We thank the reviewer and the editor for the insightful and useful comments and these have been used to revise the manuscript accordingly. Please find stated below our point-by-point response to all the comments.

EDITOR’S COMMENTS

Response: The reference has been reviewed and found complete.

REVIEWER’S COMMENTS

Reviewer #1:

Comment-1) I am fine with the corrections made, except for the following:

I would advise against the approach you took in your rebuttal to reviewer comments. On several instances, the feeling appeared to be 'while we thank the reviewer for the comment, we did it the right way' and will disregard it.

Response-1: We are truly grateful for the valuable comments offered by the reviewer. We sincerely apologize for any perceived disregard of some of the comments. This was not our intention and we will never set out to do that. Perhaps, our desire to provide detailed explanations underlining some of our rebuttal may have accounted for this. We value the suggested corrections proposed by the reviewer and we are convinced it can only improve the scholarly value of the manuscript.

Comment-2) For example, regarding the Likert scale sequence, it will not suffice to say that responders were 'given detailed instructions' and things were successful. Likert scales are very tricky devices, and to ask for input into what came b/n Low and Full was a reasonable request to make.

Response 2: We totally concur with the reviewer that the request for explanation of the response scale used for the self-perceived competency assessment is reasonable. We did not disregard this request but just tried to explain how it was done as stated in the manuscript. In fact, the scale that we used in the study was not a Likert scale but a semantic differential scale which allowed respondents to choose from paired adjectives at the opposite end of a continuum (1 to 10) (1: low competence, 10: full competence). This has been made clear in the revised manuscript [line 175].

“The participants ranked their responses to the 12 competency elements (condition- and non-condition specific) in Section B on a semantic differential scale of 1 to 10 (1: low competence, 10: full competence)”.

Comment-3) Secondly, to state that Description and Signs/Symptoms were sufficiently distinct items, I would love to be enlightened. When I think of the Description of the Common Cold, I conjure up "a viral upper respiratory infection manifesting as nasal congestion, runny nose, perhaps a cough, maybe a headache etc. Now, how do signs/symptoms deviate from that?

Response-3: We thank the reviewer and concur that there are possibility of overlap between description and sign/symptoms. However, we are of the opinion that this does not obviate the difference between both items. We thought that a description/definition is generally an overview or broad characterization while signs/symptoms are usually more specific. For instance, a viral upper respiratory infection can be described as infection of the upper respiratory tract caused by common cold virus such as a rhinovirus, influenza virus or coronavirus. However, we concur that some people may proceed to add sign and symptoms to the definition / description. 

Comment-4) For line 346, the statement -- "probably suggests the study findings are valid and reliable" -- based on a sole Cronbach's alpha is too far-reaching. I like your items in Table 3. Regardless of that stance, do you really feel you got things so right in the survey to make that conclusion? I look back to all the OTC research I have done over 30 years and I STILL worry about what mistakes I made.

Response-4: We concur with the reviewer that the risk of an over-reach is always an issue in survey research. The manuscript has been revised to ‘tone down’ the phrase to “probably suggest that the study findings are valid” [line 346].

Comment-5) Lastly, to downgrade "significantly add to the literature" to "may significantly add" is still clearly over-valuing the work you created. It is just not that profound, similar to the rest of us working in this area.

Response-5: We thank the reviewer for this suggestion. The manuscript has been revised as follow: “may add to global knowledge in the study area” [Line 339].

---

## [Editor Report · Decision Letter 2]

2 Aug 2021

Management of common minor ailments in Qatar: Community pharmacists’ self-perceived competency and its predictors

PONE-D-21-08313R2

Dear Dr. Yusuff,

We’re pleased to inform you that your manuscript has been judged scientifically suitable for publication and will be formally accepted for publication once it meets all outstanding technical requirements.

Kind regards,

Jenny Wilkinson, PhD

Academic Editor

PLOS ONE

Additional Editor Comments (optional):

Thank you for your responses, these have addressed the reviewer comments
---

## [Editor Report · Acceptance letter]

6 Aug 2021

PONE-D-21-08313R2 

Management of common minor ailments in Qatar: Community pharmacists’ self-perceived competency and its predictors 

Dear Dr. Yusuff:

I'm pleased to inform you that your manuscript has been deemed suitable for publication in PLOS ONE. Congratulations! Your manuscript is now with our production department. 

Kind regards, 

on behalf of

Dr Jenny Wilkinson 

Academic Editor

PLOS ONE